# Nb_2_O_5_ Coating to Improve the Cyclic Stability and Voltage Decay of Li-Rich Cathode Material for Lithium-Ion Battery

**DOI:** 10.3390/molecules28093890

**Published:** 2023-05-05

**Authors:** Yanlin Liu, Ruifeng Yang, Xinxi Li, Wensheng Yang, Yuanwei Lin, Guoqing Zhang, Lijuan Wang

**Affiliations:** 1School of Material and Energy, Guangdong University of Technology, Guangzhou 510006, China; 2School of Automobile and Transportation Engineering, Guangdong Polytechnic Normal University, Guangzhou 510640, China; 3College of Petroleum and Chemical Technology, Liaoning Petrochemical University, Fushun 113001, China

**Keywords:** Li-ion batteries, Li- and Mn-rich layered oxide cathodes, Nb_2_O_5_ coating, capacity, voltage decay

## Abstract

The commercialization of lithium manganese oxide (LMO) is seriously hindered by several drawbacks, such as low initial Coulombic efficiency, the degradation of the voltage and capacity during cycling, and the poor rating performance. Developing a simple and scalable synthesis for engineering with surface coating layers is significant and challenging for the commercial prospects of LMO oxides. Herein, we have proposed an efficient engineering strategy with a Nb_2_O_5_ coating layer. We dissolved niobate (V) ammonium oxalate hydrate and stoichiometric rich LMO (RLM) in deionized water and stirred constantly. Then, the target product was calcined at high temperature. The discharge capacity of the Nb_2_O_5_ coating RLM is increased from 195 mAh·g^−1^ (the RLM without Nb_2_O_5_) to 215 mAh·g^−1^ at a coating volume ratio of 0.010. The average voltage decay was 4.38 mV/cycle, which was far lower than the 7.50 mV/cycle for the pure LMO. The electrochemical kinetics results indicated that the performance was superior with the buffer engineering by the Nb_2_O_5_ coating of RLM, which provided an excellent lithium-ion conduction channel, and improved diffusion kinetics, capacity fading, and voltage decay. This reveals the strong potential of the Nb_2_O_5_ coating in the field of cathode materials for lithium-ion batteries.

## 1. Introduction

Lithium-ion batteries have been widely utilized in various kinds of electronic devices due to their high energy density and long cycling life. Compared with the anode, the cathode material has a lower specific capacity, resulting in a low energy density for advanced devices [1]. Among the recently researched cathodes, the layered Li-rich oxide materials, such as xLi_2_MnO_3_·(1 − x)LiMO_2_ (M = Mn, Ni, Co) (LLMO), have become potential materials to meet the increasing demand for energy density because of their large discharge-specific capacities from 280 to 310 mAhg^−1^ and high working potential of 3.7 V (vs. Li/Li^+^). However, there are still some drawbacks to be addressed, such as the cyclic instability, poor rate, fast capacity/voltage decay, and low initial Coulombic efficiency, which limit their large-scale commercial application [2,3,4].

The mechanism of the capacity fade and voltage decay for the LLMO has attracted attention in recent decades [2,3]. An anion redox reaction occurs during the charge and discharge of the LLMO, along with the transformation of the reactant and the intermediate reaction, such as O^2−^→O_2_^n−^ and O^2−^→O_2_, and the latter reaction is irreversible [5,6,7,8]. The irreversible reaction worsens the electrochemical performance of the LLMO in terms of the electrolyte decomposition and structural reorganization. The continuous electrolyte decomposition on the electrode−electrolyte interface results in the formation of a solid electrolyte interface (SEI) film with high resistance [9,10,11]. The transition metal ions migrate into the Li layer and then induce the phase transformation from the layered to the LiMn_2_O_4_ spinel-like structure due to the irreversible release of O_2_ on the surface. Thus, the LLMO suffers poor cycling stability.

Surface modification has been regarded as an effective strategy to improve the interface stability and thus the electrochemical performance of the LLMO. The surface coating layers usually include three categories. One is an inert material coating, such as ZrO_2_, SnO_2_, AlF_3_, and LiF [12,13,14], which can isolate the electrode and electrolyte and then prevent the electrolyte decomposing and the electrode dissolving. However, a large resistance will be introduced because the coating materials are insulators. Thus, ionic conductors and electric conductors, such as LiAlO_2_, LiPO_4_, and C, are widely used as coating materials to resolve the accompanying increased resistance [15,16,17,18]. Among them, the coating of electronic conductors, such as graphene, can increase the conductivity of the material and improve the efficiency of electron transmission. A spinel, for example Li_4_Mn_5_O_12_, is one of the most popular ionic-conductor coating materials, which can not only build an effective electrode−electrolyte interface (EEI) but also provide good Li^+^ transmission channels and thus good dynamics [19,20]. Additionally, the spinel is still an excellent buffer that can suppress the Mn-based Li-rich material transformation from a layered to a spinel structure, which was also regarded as the main reason for the fast voltage decay of the LLMO. However, the Mn dissolution, homogeneity, appropriate thickness, and the complex preparation technique are still challenges for spinel coating [21,22,23]. Recently, the concept of double-compound coating has been adopted to improve the cyclic life of LLMOs. For example, an improvement in cycling life can be obtained by using a Li_4_Mn_5_O_12_ and MgF_2_ double-compound as a coating layer, in which Li_4_Mn_5_O_12_ was used as the Li^+^ transmission channels and MgF_2_ was used as the separator between the electrode and electrolyte [24]. Nevertheless, the resistance still increases owing to the insulation of MgF_2_. 

Nb_2_O_5_ has been widely utilized in the field of biomaterials and lithium-ion batteries owing to its strong corrosive resistance, thermal stability, and high Li^+^ conductivity [25,26,27]. The effects of the Nb_2_O_5_ coating on the cycling life and voltage decay of LLMO have been reported. Wook et al. studied the Nb_2_O_5_ coating layer on the LLMO, and Pan et al. further studied the mechanism of the improved cycling life and voltage stability [28,29]. They proposed that the Li^+^ conductor and the inhabitation of the LLMO in a spinel were the main reasons for the improvement. Nevertheless, the process of Nb_2_O_5_ on the LLMO is still not clear. Herein, Nb_2_O_5_ was coated on the Li-rich material (RLM-1) to investigate the mechanism of the spinel formation and protection. Then, a new method was proposed to explain the real roles of Nb_2_O_5_ during the improved and stable charge and discharge process, which is different from Pan’s one [29]. This discovery may greatly benefit the developing of a more stable Li-rich cathode material. 

## 2. Results and Discussion

In Figure 1, it shows the initial charge−discharge curves that were the initial discharge capacities and Coulombic efficiency and d*Q*/d*V* curves for RLM-1 coated with different amounts of Nb_2_O_5_. All the charge curves consist of the slash before 4.5 V and the platform ca. 4.5 V, corresponding to the oxidation of the transition metal ion, M^3+^→M^4+^, and the activation of the Li_2_MnO_3_ component, respectively. The latter reaction along with the transfer of Mn and the oxidation of crystal oxygen, O^2−^→O_2_^n−^ (n = 1, 2, 3…) is an irreversible process [5,7,8], which leads to the low initial Coulombic efficiency for the Li-rich material. It is noted that the low Coulombic efficiency will also be exacerbated by the irreversible decomposition of the electrolyte, which is caused by the oxygen anion attack on the electrolyte, which showed 69.8% for the blank RLM-1. It is worth noting that a discharge platform is presented in 3.6 V, which is found in the discharge curve of RLM-1, convincingly suggesting the spinel component is involved in the electrochemical reaction process of the first cycle.

Comparatively, similar charge−discharge curves appeared for the Nb_2_O_5_-coated RLM-1 (RLM@Nb_2_O_5_), which indicated that Nb_2_O_5_ did not change the bulk structure of the RLM-1. When the Nb_2_O_5_ coating layer was introduced, the discharge capacity gradually increased. Figure 1A−G shows that the discharge capacity increased from 195 mAh·g^−1^ for the RLM without Nb_2_O_5_ to 215 mAh·g^−1^, which reaches the maximum when n(Nb): n(M) = 0.010. However, when the ratio amount of the Nb_2_O_5_ coating layer increased to 0.050, the discharge capacity drops to 184 mAh·g^−1^, and this dropping trend continues as it increases. To investigate the functional mechanism of Nb_2_O_5_ on the enhanced discharge capacity, the d*Q*/d*V* curves of each sample were made, as shown in Figure 1I,J. It is observed that the coating layer shows no obvious effect on the position of the voltage peak and current. The peaks are located between 2.7 and 3.0 V and correspond to the reduction in Mn^4+^ in the spinel phase in the lamellar structure. Therefore, the discharge current appears in the range of 2.75–3.0 V, which is regarded as belonging to the spinel structure formed by the transfer of Mn, and becomes larger as the Nb_2_O_5_ amount increases [29]. Additionally, the current peak around 3.7 V shows a blue shift when the Nb_2_O_5_ increases, which may due to the introduced resistance by the too-thick Nb_2_O_5_ layer. Specifically, the initial Coulombic efficiency shows an obvious increasing trend that accompanies the increase in Nb_2_O_5_, and when the Nb: M was 0.010, it reached to 84.4%, which is much higher than others. Thus, it can be seen that the Nb_2_O_5_ can reduce the irreversible charge capacity through enhancing the activation of the Li_2_MnO_3_ component. 

Figure 2A shows the cyclic performance for each kind of coating amount of Nb_2_O_5_. We observed that the blank RLM-1 keeps good capacity retention at the beginning of eight cycles. The initial capacity was raised when the Nb_2_O_5_ coating layer was introduced when coating ratio was lower than 0.025, and the capacity increased as the ratio began to increase, with the trend remaining steady for the following cycles. It is suggested that the Li_2_MnO_3_ component can be fully activated by Nb_2_O_5_ coating along with an infiltrating process by the electrolyte. However, the initial capacity begins to fall when the coating ratio exceeds 0.05, which was expected, as the high resistance was also introduced when large amounts of Nb_2_O_5_ were used to further investigate the cycling-voltage curves, as shown in Figure 2B. It can be found that the initial voltage and the amount of Nb_2_O_5_ show an anti-correlation relationship. The initial voltage becomes lower, but the voltage decay becomes slower. The average voltage decay for the coating ratio of 0.010 is 4.38 mV/cycle, which is far lower than the 7.50 mV/cycle for the blank one. Thus, it is suggested that the Nb_2_O_5_ coating is beneficial for easing the voltage decay of Li-rich material; therefore, the optimal coating ratio is 0.010. 

The dQ/dV curves of the eighth circle for each group are shown in Figure 2C,D. All of them have a current charge after 4.4 V, even if it is at the 8th circle, indicating that they have reversibility to a certain extent above 4.4 V, which has also been reported in the literature. The amount of coating has no obvious influence on the charging voltage peak and current. It is worth noting that there is a spinel discharge current around 2.75–3.0 V, which does not exist in the first cycle for the blank RLM-1, suggesting some phase transformations occur [29]. Comparatively, the spinel discharge current becomes obvious when the Nb_2_O_5_ is used, even if the coating ratio is only 0.005, which means that this coating can effectively accelerate the formation of the spinel phase, the amount of which is highly positively correlated with the ratios of the introduced Nb_2_O_5_. This may be due to the changed dynamic processes, which were related to the topotaxial factors, such as the migration channels of Li^+^, Mn, and O for the phase transformation of Li_2_MnO_3_ to spinel [30]. There are no obvious differences among all the samples for the coating ratio below 0.050. Moreover, the spinel current also shows little change compared with the first-cycle discharge, as shown in Figure 1J, when the coating ratio was at the range of 0.010–0.025, suggesting that the spinel nearly formed less at the following cycles. This explains the lower initial voltage and the slower voltage decay for the Nb_2_O_5_ coating material, and also agrees with the known mechanism of voltage decay for Li-rich material, namely, formed spinel. Notably, a larger response current below 3.6 V for the 0.050 and 0.075 coating ratios is observed, and this can be ascribed to the excessive spinel phase formed by the increasing amount of Nb_2_O_5_. Moreover, the huge decrease in the layered structural component was implicated in the excessive spinel phase forming, resulting in the smaller current in the range of 3.8–4.6 V. The obviously positive shift of the charging peaks indicates the larger polarization caused by the excessive Nb_2_O_5_ when the coating ratio is above 0.050.

Figure 3 displays the cycling performance for each sample with Nb/M 0.010 calcined with 550 °C, 600 °C, 650 °C, 700 °C, 750 °C, 800 °C, 850 °C, and 900 °C. The sample delivery is 195 mAh·g^−1^ when the calcined temperature is 550 °C, indicating that the low temperature has no influence on the discharge capacity. It is raised as the temperature increased, and it reaches the highest level of 215 mAh·g^−1^ when the temperature is at 850 °C. The initial Coulombic efficiency shows a similar trend, and it reaches 78.7% at the calcined temperature of 800 °C. Although the discharge platform shows a downward trend with the increase in calcination temperature, the decay rate gradually decreases and the average voltage decay is 11.49 mV/cycle for 550 °C, while it is only 7.76 mV/cycle for 850 °C for the first 8 cycles. Combining the initial charge−discharge curves and the dQ/dV differential curves, as shown in Figure 3C−E, on the one hand, it can be found that the temperature increase accompanies the polarization of the electrode, as the first oxidation potential shows a slight positive shift. On the other hand, although each sample has discharge peaks at 4.3 V, 3.75 V, and 3.0 V, respectively, the current of the spinel phase corresponds to the peak at around 3.0 V for a temperature of 550 °C, which is slightly lower than other samples. Additionally, the current shows a slight increase as the temperature trend continues upward. This can be attributed to the better crystallization of Nb_2_O_5_ under the higher calcined temperature, which provides an excellent transmitting channel for oxygen anions, the Mn, and Li^+^ to form spinel, as well as creating a small discharge polarization. However, the increase in temperature also increases the particle size of Nb_2_O_5_ crystals, which can be found in the Appendix A, which leads to the long Li^+^ transmission distance and large polarization. Thus, it shows the decreased discharge capacity and the low Coulombic efficiency. Our analysis shows that the 850 °C calcination temperature for the samples is the optimum, which is suggested to be utilized in future research. 

Figure 4A shows the XRD patterns of the RLM-1 and RLM@Nb_2_O_5_ materials with a coating ratio of 0.010 calcined at 850 °C. In both of them, there is a splitting peak at 20–23°, corresponding to the Li/Mn cations mixed in the monoclinic C2/m Li_2_MnO_3_ region, which are considered to be characteristic of the diffraction peaks of Li-rich materials [3,31]. The Nb_2_O_5_ coating has no influence on the diffraction peaks of RLM-1, indicating that the Nb_2_O_5_ coating layer does not change the main lattice of the RLM-1. The peaks at 42.9° and 62.3° correspond to the (200) and (220) crystal planes of NbO, re-spectively. The peak intensity ratios of RLM@Nb_2_O_5_(101)/(003) were 0.230 and (004)/(003) were 0.539, while the peak intensities of (101)/(003) and (004)/(003) of RLM-1 were 0.2612 and 0.603, respectively. The half-peak width of the main peak of the sample with Nb_2_O_5_ compared with RLM-1 was 1.031. The half-widths of the peaks are hardly changed, so Nb_2_O_5_ does not change the main lattice of RLM-1. Additionally, the content of Nb_2_O_5_ is small, and only extremely weak diffraction peaks of Nb_2_O_5_ are detected in the XRD diffraction patterns around 43° and 62°. In addition, as shown in Figure 4B, it can be observed that the 3d3/2, 3d5/2, and 3d5/2 peaks in photoelectron spectroscopy (XPS) are located at 209.9, 207.3 and 207.1 ev, respectively, which belong to Nb_2_O_5_ [32,33]. This indicates that Nb2O5 had been effectively coated on the surface of RLM-1.

Figure 5 shows the element mapping of RLM-1 and RLM@Nb_2_O_5_ materials, and it could be observed that the Ni, Mn, and O elements are distributed very uniformly before and after the coating, indicating that the coating has no influence on the homogeneous nature of each element inside the material. Moreover, the signal of the Nb element can be observed in the RLM@Nb_2_O_5_ material. However, its signal intensity is weak, which may be attributed to the low concentration of Nb, as shown in Figure 5G, indicating that the Nb_2_O_5_-coated layer is even. 

Figure 6A shows the improvement in the Nb_2_O_5_ coating for RLM-1. The blank RLM-1 shows a serious degradation at 170 cycles, and the capacity is 83.1 mAhg^−1^, only about 42.1% of the initial capacity at 300 cycles. Comparing with RLM-1, the RLM@Nb_2_O_5_ shows good cycling performance with a capacity retention of 81.5% at 300 cycles. This suggests that the proper coating proportion with Nb_2_O_5_ can effectively inhibit the capacity decay of a lithium-rich cathode. It is found that the voltage platform trend presents a similar behavior, which reveals a fast decay at 170 cycles for blank RLM-1 while retaining a good stability for RLM@Nb_2_O_5_, as shown in Figure 6B. The voltage platform retention is calculated by the Formula (1):(1)$$\eta =\frac{V_{\mathrm{cycle}}}{V_{\mathrm{initial}}}\times 100\%$$
where η represents the voltage platform retention; V_initial_ represents the discharge voltage plateau of the first cycle; V_cycle_ represents the discharge voltage plateau for the corresponding number of cycles.

The discharge voltage plateau of RLM-1 dropped from the initial 3.8604 V to 3.3664 V, with a voltage retention of 87.2% and average decay of 1.65 mV/cycle for 300 cycles. After coating with Nb_2_O_5_, the voltage decay was eased from 3.7709 V to 3.5789 V, with a voltage retention of 94.9% and average decay of 0.64 mV/cycle. This indicates that Nb_2_O_5_ coating can also effectively suppress the voltage decay of Li-rich materials. It is worth noting that the RLM@Nb_2_O_5_ shows a large capacity fade at 210 cycles, while there is no corresponding accelerated voltage decay, indicating that there is no certain relationship between the capacity fade and voltage decay, which is utilized to explain the real reason for the long-time voltage decay of lithium-rich materials. 

From the dQ/dV analysis, as seen Figure 6C, it can be found that both RLM-1 and RLM@Nb_2_O_5_ show similar electrochemical behaviors, namely, charging peaks between 3.5 and 3.75 V and broad discharge peaks around 4.1 V and 3.0–3.6 V before 150 cycles. There are small charging peaks of about 4.5 V for RLM-1 at the 150th cycle, which indicates that the activated Li_2_MnO_3_ structures still lack wrapping. The RLM-1 shows a large polarization and irreversibility with the charge current peak between 3.5 and 3.75 V, and the discharge current above 3.7 V completely disappeared after 300 cycles, while it kept the same electrochemical performance with 150 cycles for RLM@Nb_2_O_5_. Figure 6D shows the impedance diagram of two samples after 300 cycles. The entire impedance diagram is composed of two semicircles in the middle- and high-frequency regions, indicating the solid electrolyte−electrode interface film (abbreviated as EEI) has been formed on the surface of the electrode. It can be found that the solution impedance R_s_ of the two samples is nearly equivalent. The R_EEI_ of the RLM-1 in the high-frequency region is greater than that of RLM@Nb_2_O_5_. This may be due to the thicker poor-conductivity film formed by the deposition of electrolyte decomposition on the surface of the RLM-1 material. Moreover, the ion conductivity of Nb_2_O_5_ coating also reduces the resistance of the film. In the mid-frequency region, the impedance semicircle radius of RLM-1 is smaller than that of RLM@Nb_2_O_5_, indicating a lower charge transfer resistance R_ct_ of RLM-1 than that of RLM@Nb_2_O_5_. This may be due to the reduced size of the Li^+^ transmission distance by the powdering process of material particles during cycling, while Nb_2_O_5_-coated RLM-1 maintains particle integration and also shows higher R_ct_. The parameters are shown in Table 1. As mentioned above, the better cyclic stability and lower voltage decay originates from the Nb_2_O_5_ coating layer’s modified surface.

The TEM images of RLM-1 and Nb_2_O_5_ RLM before and after 300 cycles are presented in Figure 7. It can be found from Figure 7A that the original RLM-1 was covered by a homogeneous coating, a dense and amorphous layer with about 4 nm thickness. The thickness of the original covered layer has no change after secondary Nb_2_O_5_ coating; however, some integral crystalline Nb_2_O_5_ was observed and passed through the covered layer, as shown in Figure 7B. This can provide good channels for Li^+^- and O^2-^-exchange between the electrode material and electrolyte because of the good lattice as well as the ion conductivity. After 300 cycles, the RLM-1 particles were completely exposed in the electrolyte and broken up, with the original covered layer completely destroyed, as presented in Figure 8C. This also explains the phenomenon of the fast capacity fade and voltage decay for RLM-1. It indicated that RLM-1 cannot be protected effectively by the original amorphous layer. Comparatively, the original covered layer still exists and has no obvious change after 300 cycles for the RLM@Nb_2_O_5_. Remarkably, comparing with the RLM-1, a new spinel layer was formed between the original cover and the bulk material for the RLM@Nb_2_O_5_. This can be explained as the Nb_2_O_5_ crystal providing good channels for Li^+^ and O^2−^ exchange, resulting in a lower oxygen anion, which is beneficial for forming the spinel structure during the Li_2_MnO_3_ activation process. Moreover, the high-concentration spinel-phase structure not only provides the Li^+^ channels [19,20] but also inhibits the further forming of spinel, which causes the lower initial discharge voltage platform and the smaller voltage decay for the RLM@Nb_2_O_5_. 

The SEM images show similar results as above, shown in Figure 8. It can be clearly seen that the particle sizes of RLM-1 and RLM@Nb_2_O_5_ were about 10 μm, stacking with lots of smooth and smaller particles, indicating that Nb_2_O_5_ coating has no influence on the topography of RLM-1. However, the boundary between the particles becomes fuzzy and covered by lots of electrolyte decomposition after 300 cycles for RLM-1. Comparatively, the morphology of RLM@Nb_2_O_5_ has no obvious change. Especially, it does not exhibit the electrolyte decomposition, which indicates that Nb_2_O_5_ secondary coating can suppress the decomposition of electrolytes. 

XPS spectra were conducted to analyze the surface compositions of Li-rich electrodes after 300 cycles. As shown in Figure 9, in the C 1s spectrum, the peak at 284.3 eV is ascribed to the conductive carbon [32]. The peaks at 286.5 eV, 288.6 eV, and 290.8 eV are assigned to the C-O, C=O, and C-F groups of ROCO_2_Li, ROLi, and PVDF species, respectively, which result from the electrolyte decomposition [32,34]. Considering the larger C-O intensity of RLM-1 than that of RLM@Nb_2_O_5_, it should be concluded that a lot of electrolytes were decomposed and deposited on the surface of the RLM-1 electrode.

In the O 1s spectrum, the peaks at 531.3–531.5 eV (C=O, Li-O), 533–534 eV (C-O), and 529.8 eV (M-O) are characteristic of polycarbonates, lithium alkyl carbonates, and metal oxide, respectively [35]. The weaker intensity of these peaks for the RLM@Nb_2_O_5_ electrode compared with RLM-1 suggests that more electrode was exposed due to the surface coating layer being damaged. It should be noticed that the peak intensities between 531.3 and 531.5 eV are the multiple peaks for the C=O of polycarbonates and Li-O of Li_2_O. According to the C 1s spectrum, the C=O peak of RLM@Nb_2_O_5_ should be smaller than that of RLM-1 because less electrolyte being decomposed. Interestingly, the peak intensity in the range of 531.3–531.5 eV for RLM@Nb_2_O_5_ is larger than that for RLM-1, indicating that there is more Li_2_O on the surface of RLM@Nb_2_O_5_ [35,36,37,38,39]. The Li_2_O was formed by the reaction of O^2−^ and Li^+^ during Li_2_MnO_3_ activation, which can be scripted as 5Li_2_MnO_3_ → Li_4_Mn_5_O_12_ + 3Li_2_O, which produced Li_2_O and the spinel structure simultaneously. This result corresponds with the description of the spinel behavior mentioned above, since the local structure changes to the spinel structure in the Li_2_MnO_3_ component are caused by Li+ extraction, which was confirmed by Yu’s group [40,41,42]. The larger spinel structure and Li_2_O can account for the accelerated O and Li^+^ extraction from the Li_2_MnO_3_ component, as well as the faster dynamic processes induced by the lower topotaxial factor of O and Li^+^ from the better O and Li^+^ channels that are provided by Nb_2_O_5_ coating layer [19,20,30]. The view of the above-mentioned accelerated spinel-formation process is different from the one that observed by Pan et al. [29]. In the F 1s spectra, the peaks of 687.5 eV, 686.2 eV, and 685.0 eV correspond to PVDF, Li_x_PO_y_F_z_, and LiF, respectively [39]. They produced a small amount of LiF through the reaction with HF in contact with the electrode, meaning they have a critical role in maintaining the fasting Li-ion diffusion kinetics and the wonderful electrochemical performance of the electrode. The weaker peak intensity of Li_x_PO_y_F_z_ for the RLM@Nb_2_O_5_ electrode suggests less electrolyte decomposition [34]. The Nb signal of the RLM@Nb_2_O_5_ electrode after 300 cycles further confirms that Nb_2_O_5_ plays a role across the whole cycling life. 

## 3. Experiment Section

### 3.1. Preparation of Nb_2_O_5_ Coated Samples

An appropriate amount of ammonium niobate (V) oxalate hydrate was dissolved in deionized water and then constantly stirred for 30 min. After the stoichiometric RLM-1 was added into the solution above, the mixture was constantly stirred under room temperature for 3 h and then heated at 75 °C. The molar ratios of Nb to M (M = Ni, Co, Mn) are 0.005, 0.010, 0.020, 0.025, 0.050, and 0.075, respectively. The obtained powder was dried at 100 °C for 12 h in vacuum, preheated at 500 °C for 6 h, and then calcinated at 900 °C for 12 h with 2 °C/min. RLM-1 was purchased from Jiangxi Jiangte Lithium-Ion Battery Material Co. (Chaoxia Road, Pharmaceutical Industrial Park, Yuanzhou District, Yichun City, Jiangxi Province, China), Ltd. In addition, the calcining temperatures were also studied from 550 to 900 °C.

### 3.2. Physical Characterizations

The crystalline structures of the resulting samples were characterized using an X-ray diffractometer (XRD, D8 Advance, Bruker, Germany) in the 2θ range of 10–80° at an interval of 0.2 °/min. The particle sizes were measured by a laser particle-size analyzer (BT-9300HT, SuZhou Hinos Industrial Co., China). The morphologies and nano structures were observed using scanning electron microscopy (SEM, ZEISS Ultra 55, Zeiss, Germany) and transmission electron microscopy (TEM, JEM-2100HR, JEOL, Japan), respectively. X-ray photoelectron spectroscopies (XPS) were conducted on a Thermo Scientific TM K-Alpha TM^+^ spectrometer equipped with a monochromatic Al Kα X-ray source (1486.6 eV) operating at 100 W. Samples were analyzed under vacuum (*p* < 10^−8^ mbar) with a pass energy of 150 eV (survey scans) or 25 eV (high-resolution scans). All the peaks were calibrated with C1s peak binding energy at 284.3 eV. The experimental peaks were fitted with XPSPEAK41 software.

### 3.3. Electrochemical Measurements

For electrochemical measurements, the electrodes were prepared by coating the slurry including 93 wt.% active material, 4 wt.% conductive carbon (Super P), and 3 wt.% binder (polyvinylidene fluoride) onto Al foils. CR2032 coin cells were assembled in a glove box filled with highly pure Ar using the prepared electrode as the working electrode and Li metal as the counter electrode. In total, 1.0 mol dm^−3^ LiPF_6_ in ethylene carbonate (EC)/ethyl methyl carbonate (EMC)/diethyl carbonate (DEC) (1:1:1 wt.%) (Dongguan Kaixin Battery Material Co., Ltd., Dongguan, China) was used as the electrolyte, and we used a microporous membrane of 16 μm (purchased from Shenzhen, China, Senior Technology Material Co, Ltd.) as the separator. Charge−discharge tests were carried out on a land testing system in the range of 2.0–4.8 V. The electrochemical impedance spectroscopy measurements were carried out at 3.15 V at the specific cycle on the Autolab PGSTAT302 from 100 KHz to 0.01 Hz with 5 mV amplitude. 

## 4. Conclusions

The Nb_2_O_5_ secondary coating can effectively improve the first Coulombic efficiency, capacity fade, and voltage decay of the commercial lithium-rich manganese-based cathode. A simple amorphous coating layer cannot protect the Li-rich material effectively for a long cycling life. The Nb_2_O_5_-coated Li-rich materials with low resistance, high capacity, and voltage stability can be obtained when the coating ratio is 0.010 and the calcining temperature is 850 °C. Nb_2_O_5_ with a high crystalline structure can provide excellent oxygen anion and Li^+^ transport channels during Li_2_MnO_3_ activation and then accelerate the spinel formed at the coating region to build a stable and highly dynamic coating layer [19,20]. The spinel formed at an early stage can suppress the formation in the following cycles and ease the voltage decay during cycling. Moreover, the inert Nb_2_O_5_ coating layer can effectively protect the electrode material from corrosion and inhibit the oxidation and decomposition of the electrolyte.

## Figures and Tables

**Figure 1 molecules-28-03890-f001:**
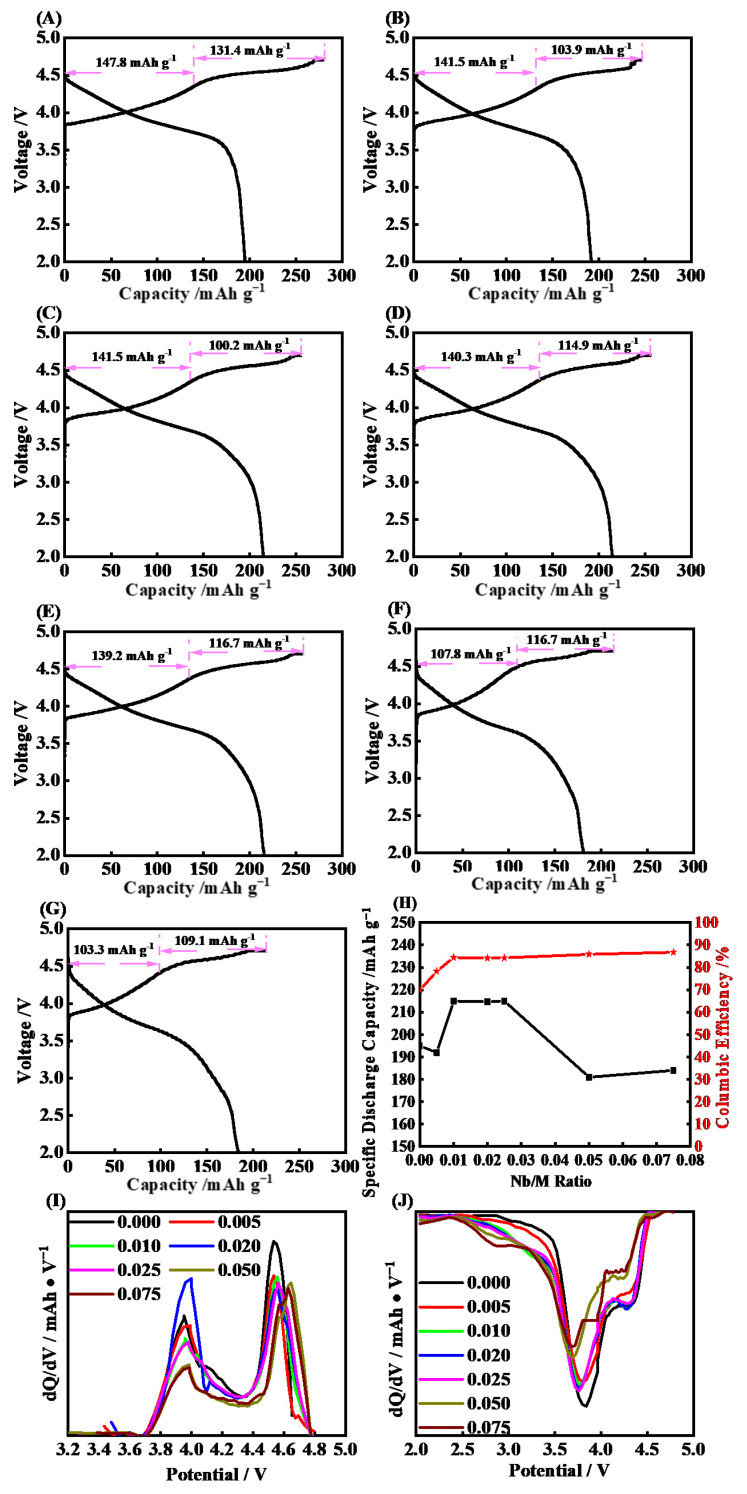
The initial charge−discharge curves (**A**−**G**), discharge capacity and Coulombic efficiency (**H**), and dQ/dV curves for different ratios of Nb:M (**I**,**J**).

**Figure 2 molecules-28-03890-f002:**
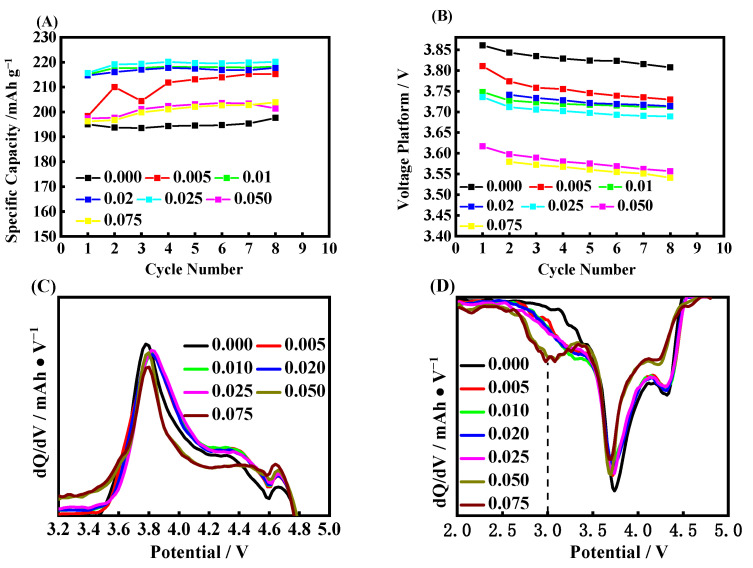
The cycling life curves (**A**), the platform of discharge (**B**), and dQ/dV curves (**C**,**D**) of the 8th cycle for different coating ratios of RLM@Nb_2_O_5_.

**Figure 3 molecules-28-03890-f003:**
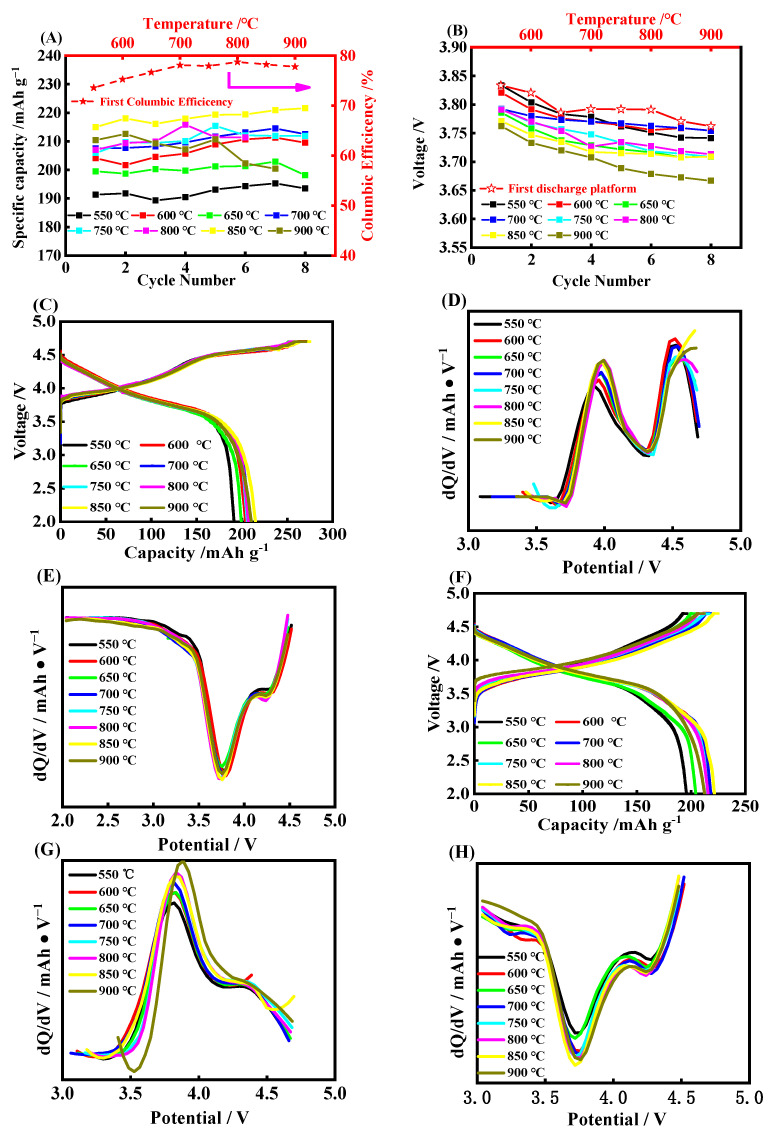
The cycling life curves (**A**), platform of discharge (**B**), charge and discharge curves and dQ/dV curves of the 1st cycle (**C**−**E**) and 8th cycle (**F**−**H**) of the samples obtained at 550 °C, 600 °C, 650 °C, 700 °C, 750 °C, 800 °C, 850 °C, and 900 °C, respectively.

**Figure 4 molecules-28-03890-f004:**
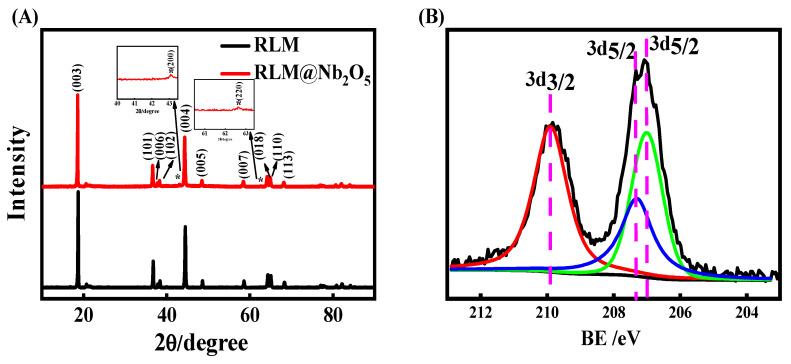
XRD patterns (**A**) and Nb XPS pattern (**B**) of RLM@Nb_2_O_5_.

**Figure 5 molecules-28-03890-f005:**
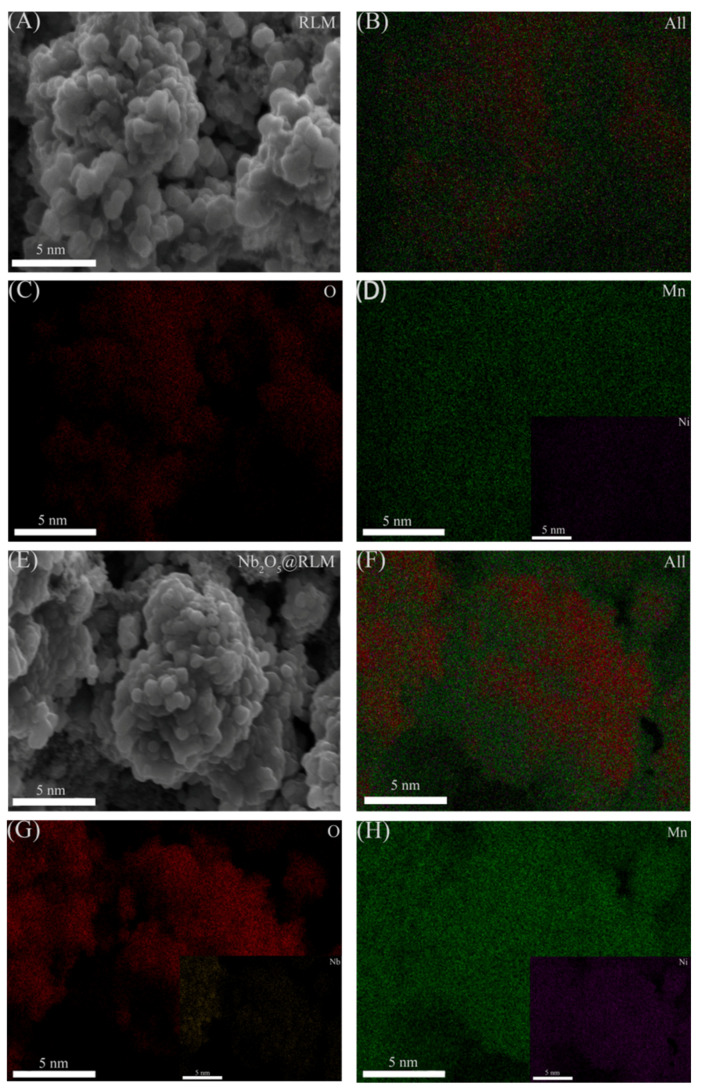
SEM images of RLM-1 (**A**), all-element mapping images of RLM-1 (**B**), O-element mapping images (**C**), Mn- and Ni-element mapping images (**D**), SEM images of RLM@Nb_2_O_5_ (**E**), all-element mapping images of RLM-1 (**F**), O- and Nb-element mapping images (**G**), Mn- and Ni-element mapping images (**H**).

**Figure 6 molecules-28-03890-f006:**
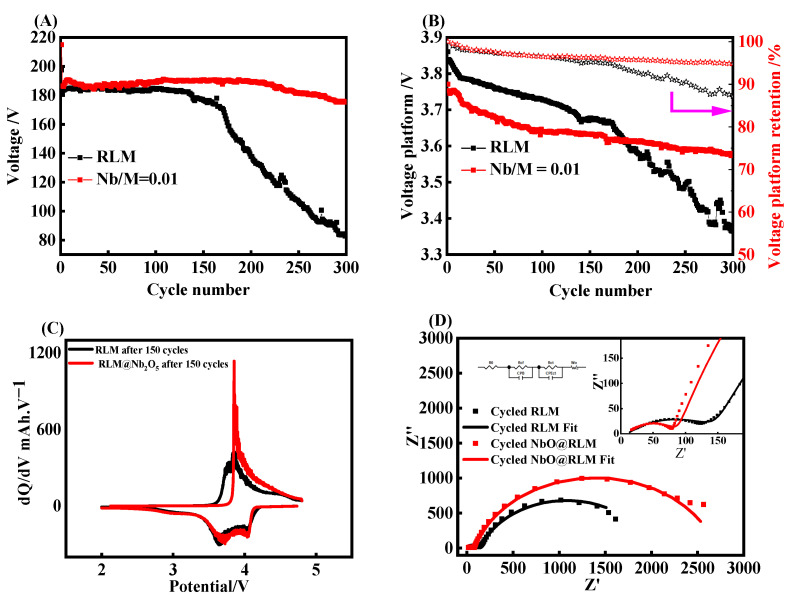
The cycling life curves (**A**), voltage platform of discharge and voltage retention (**B**), dQ/dV curves of the 150th cycle (**C**), and impedance diagram after 300 cycles (**D**).

**Figure 7 molecules-28-03890-f007:**
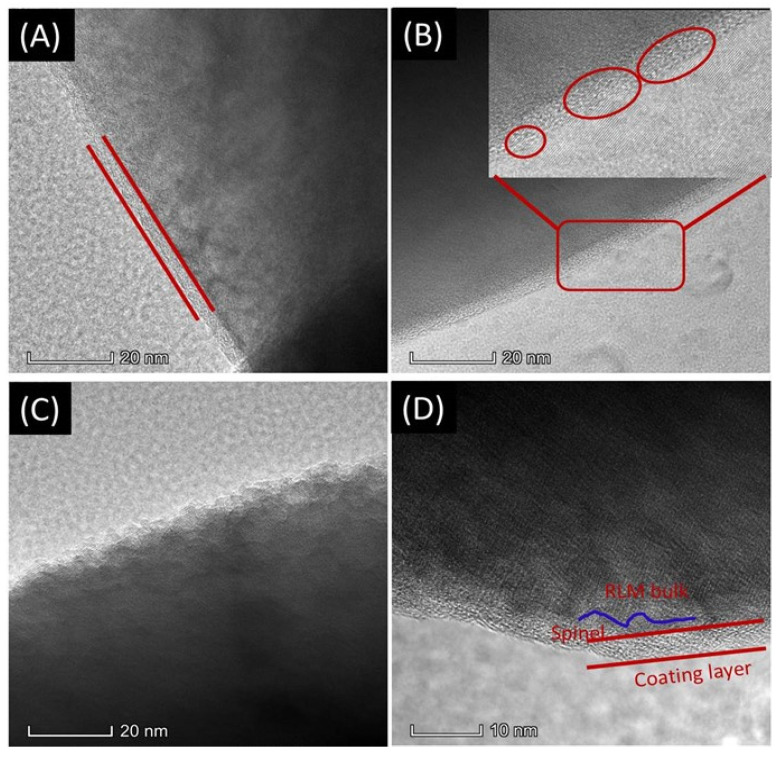
TEM of RLM-1 (**A**) and RLM@Nb_2_O_5_ (**C**), and RLM-1 (**B**) and RLM@Nb_2_O_5_ after 300 cycles (**D**).

**Figure 8 molecules-28-03890-f008:**
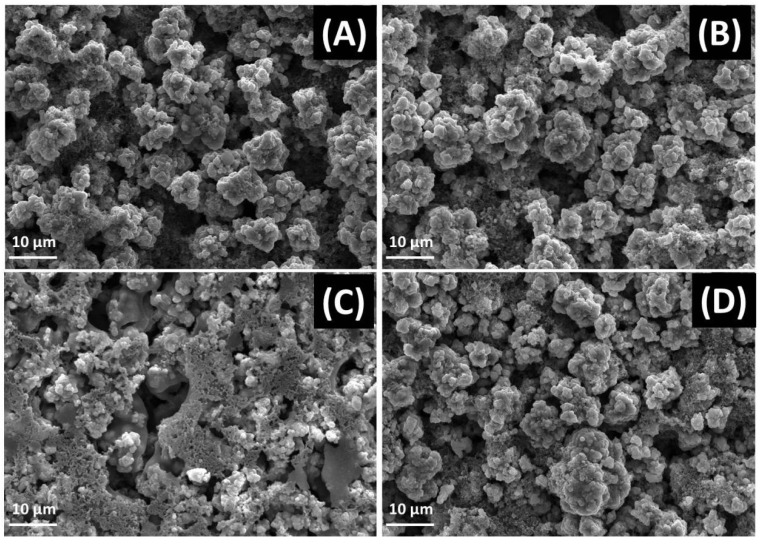
The SEM images of RLM-1 (**A**) and RLM@Nb_2_O_5_ (**B**), and RLM-1 (**C**) and RLM@Nb_2_O_5_ after 300 cycles (**D**).

**Figure 9 molecules-28-03890-f009:**
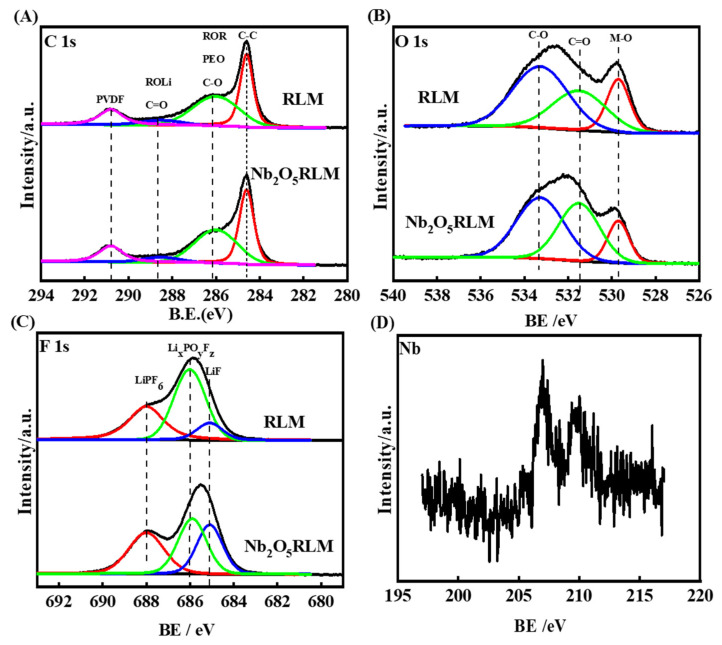
XPS spectra of RLM-1 and RLM@Nb_2_O_5_: C 1s (**A**), O 1s (**B**), F 1s (**C**), and Nb (**D**).

**Table 1 molecules-28-03890-t001:** Comparison of resistance sections of RLM and RLM@Nb_2_O_5_.

		RLM	RLM @ Nb_2_O_5_
R_s_/(Ω)		10.46	9.675
R_f_/(Ω)		143.4	80.13
CPE1	CPE1-T/(F)	1.4348 × 10^−4^	4.4665 × 10^−5^
	CPE1-P/(F)	0.46765	0.57493
R_ct_/(Ω)		1856	2610
CPE2	CPE2-T/(F)	1.9901 × 10^−2^	6.5415 × 10^−4^
	CPE2-P/(F)	0.80459	0.83649

## Data Availability

The corresponding dates have been provided in paper. And there is no new data that were created.

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
