# Peer review of "Nb2O5 Coating to Improve the Cyclic Stability and Voltage Decay of Li-Rich Cathode Material for Lithium-Ion Battery"

_molecules, 2023, doi:10.3390/molecules28093890_

Round 1

Reviewer 1 Report

The manuscript is related to the interesting scientific theme and have some new data which may be published after revision: 

1)    On Fig. 1 I, G; Fig.2 C, D; Fig. 3 G, H the peaks described in the text should be highlighted. Also, for a more visual representation, it is necessary to indicate the direction of peak replacement at different molar ratios of niobium and transition metals in the figure.

2)    When describing the peaks in Fig. 1 I, G; Fig.2 C, D, occurring in the region of 2.75-3 V, a reference should be given to the literature data, where it was shown that these peaks are responsible for the transition of the Li2MnO3phase to spinel.

3)    When describing Fig.2 C it was said that the amount of coating does not affect the curves observed during charging. However, when the element ratios are higher than 0.050, the charging curves change their shape. In the section up to 3.6 V, the current for these samples was higher than for others. At the same time, the shape of the curve also changes in sections 3.8 – 4.6 V. This phenomenon should be explained.

4)    In Fig.4 A, the peaks can be seen very indistinctly. It is recommended to provide a drawing with a high resolution. It is also necessary to put enlarged fragments of the diffractogram in order to better see the peaks from Nb2O5around 43° and 62°.

5)    When describing Fig. 4B it is necessary to provide references to the literature data, where it was shown that the peaks correspond to Nb2O5.

6)    The description of the EIS data should be reworked. It is necessary to give an equivalent scheme, as well as its parameters, which were used in modeling.

7)    During describing the results, it was said several times that the coating improves the activation of the Li2MnO3 phase, but sufficient explanations were not given. It should be explained exactly how the coating effects on activation. The assumption mentioned in the paper about the improvement of diffusion is not supported by either literary or experimental data. It is necessary to better describe the mechanism of the effect of the coating on the activation of the Li2MnO3 phase. As an example, the authors can use some of https://doi.org/10.1002/aenm.201300950, https://doi. org/10.1149/1.3151963,   https://doi.org/10.1016/j.electacta.2022.140237, or any other research they like to. 

8)    The absence of obvious signs about the coating’s formation on the surface of the cathode material. The article's synthesis method proposed the involves intensive long-term mixing of the reagents at elevated temperatures. Moreover, the cathode's and the coating's materials are directly mixed with each other. Therefore, clarity evidence, which confirms the formation of the coating on the cathode surface, instead of the doping process, is needed. Based on the results of XRD, SEM-EDX, TEM, it is possible to assume the occurrence of the second (doping) process.

9)    XRD: did the authors compare the intensities ratios of the small peaks to the largest one and did they compare the half-widths of the peaks? For example, there is a feeling that the (006) peak has disappeared on the sample with the Nb2O5 “coating” (compared to the initial sample). Also, during the initial estimation of the intensity ratios of the (101)/(003) and (004)/(003) peaks, one thing can be noticed: these ratios for the sample with Nb2O5 are greater than for the initial one, therefore, either the intensity of the (003) peak decreased for the sample with Nb2O5, or the intensities of the (101) and (004) peaks increased.

10) SEM-EDX: when comparing figures 5(D) and 5(H), it can be argued that for a sample with Nb2O5, the color intensity of Mn and Ni is greater than for the original one, if the authors did not edit the photos in the editor (gamma, saturation, contrast, etc.), then indirectly this may indicate, for example, the diffusion of Mn and Ni from the bulk to the surface. Also, the intensity of Nb in fig. 5(G) is barely distinguishable (which can be associated both with Nb low concentration and Nb presence not on the surface, but in the bulk of the sample).

11) TEM: TEM images cannot unambiguously shown the presence of the Nb2O5 crystal structure (which is described in line 210) and its presence in general, because the selected layers in figures 7(A) and 7(B) do not seem to be very different, and in the case of figure 7(B) there is no unambiguous crystal structure.

12) Some terminology is not clear: why (for example, in line 125) “n-” is a degree for the sub index, is this done intentionally or it's a mistake? Also, the designation of the sample with Nb2O5 (Nb2O5@RLM) entered in line 137 is not entirely correct, because it's hint at the presence of niobium oxide in the bulk of the cathode material (in the end, if this is a coating, then it is better to use "RLM@Nb2O5", as is usually used for fullerenes and carbon nanotubes: Li@C60 - lithium inside the fullerene, C60@Fe2O3 - iron oxide on the surface of the fullerene).

13) Line 54 mentions three categories of coating materials, while the following text highlights only two categories (either Li4Mn5O12 is the third category, but then it would be great to know its fundamental difference from the second category of coatings: LiALO2, LiPO4 and C).

Author Response

Dear Reviewer,

We have carefully revised our manuscript, according to the reviewers' very helpful comments and suggestions.

The authors are truly grateful for the kind comments and found them most helpful. We have revised the manuscript fully according to the referees’ comments and your suggestions. Details of our point-by-point responses to the referees’ comments are listed in the uploaded file. After revision, the quality of the manuscript was strongly strengthened.

Thank you very much for your support! Look forward to hearing from you.

                                                      Best regards,

Xinxi Li

Guangdong University of Technology

Reviewer 2 Report

The research described in the paper is noteworthy, and the findings presented therein have the potential to provide valuable insights for researchers working within the field. However, the overall presentation of the paper is inadequate. The manuscript contains many errors in the English language, and even Figure S2 features text written in Chinese characters. It is difficult to comprehend how such obvious issues were not addressed by the authors before submitting the manuscript for publication. Despite the potential significance of the results, the paper's numerous formal shortcomings render it unsuitable for consideration for publication in Molecules. I strongly advise the authors to thoroughly revise the manuscript and resubmit it for further evaluation.

Author Response

Dear Reviewer,

We have carefully revised our manuscript, according to the reviewers' very helpful comments and suggestions.

The authors are truly grateful for the kind comments and found them most helpful. We have revised the manuscript fully according to the referees’ comments and your suggestions. After revision, the quality of the manuscript was strongly strengthened.

Thank you very much for your support! Look forward to hearing from you.

                                                      Best regards,

Xinxi Li

Guangdong University of Technology

Round 2

Reviewer 1 Report

Dear Authors, 

Main correction done well, but still you need to fix minor errors:

- Figures 4(A), 5(E), 6(C, D), 9(A, B, C) and line 433 use “Nb2O5@RLM” instead of “RLM@Nb2O5”.

- The title of Figure 4 says XPS for bare RLM and covered, but in fact only 1 graph of XPS in figure 4.

- In the conclusion, the authors write "well crystalline structure" (line 527), although this is not proven and TEM images say otherwise.

Remark about coatings for future work, it would be good if you will study the influence of synthesis parameters on coatings thickness and conformality in your future work. For now, these aspects are not obvious. TEM is local method it is provided local thickness and conformality. By the way, 900C at 12 h can provide Nb diffusion in cathode. In future work it may be studied.

Reviewer 2 Report

The manuscript has been considerably improved, but there are still some deficiencies that need to be addressed before it can be published. The following changes need to be made:

1. There are still some English language problems. Please change the word "spectrums" to "spectra" (it is more correct to use spectra as the plural).

2. The revised version of the manuscript has comments that cannot be read (e.g., on lines 197, 430, 433, etc).

3. Line 198 (German or Germany?)

4. Please review line 226, ((n=1,2,3….) is in superscript).

5. Figures 1, 2, 3, and 6 are blurry. Please modify them appropriately.

6. It would be convenient to rephrase the sentence corresponding to lines 355 to 358...the text is confusing and difficult to understand.

7. The Figure caption for Figures 7 and 8 is confusing. Please modify them.

8. Figure 9, which corresponds to the XPS results, raises some doubts. The two peaks corresponding to Figure 9c, assigned by deconvolution to the species LixPOyFz, have different widths, which makes no sense. Please redo the deconvolution.
